# Telomere Transcription in *MLL*-Rearranged Leukemia Cell Lines: Increased Levels of TERRA Associate with Lymphoid Lineage and Are Independent of Telomere Length and Ploidy

**DOI:** 10.3390/biomedicines11030925

**Published:** 2023-03-16

**Authors:** Corrado Caslini, Amparo Serna

**Affiliations:** 1Department of Pathology, University of Michigan Medical School, Ann Arbor, MI 48109, USA; 2Department of Molecular Biology and Biochemistry, Rutgers University, Piscataway, NJ 08854, USA; 3Clinical Science Laboratory, H. Lee Moffitt Cancer Center & Research Institute, Tampa, FL 33612, USA; 4Department of Internal Medicine, University of Michigan Medical School, Ann Arbor, MI 48109, USA

**Keywords:** TERRA, telomere, transcription, MLL, acute lymphoblastic leukemia

## Abstract

Telomere transcription into telomeric repeat-containing RNA (TERRA) is an integral component of all aspects of chromosome end protection consisting of telomerase- or recombination-dependent telomere elongation, telomere capping, and the preservation of the (sub)telomeric heterochromatin structure. The chromatin modifier and transcriptional regulator MLL binds to telomeres and regulates TERRA transcription in telomere length homeostasis and response to telomere dysfunction. MLL fusion proteins (MLL-FPs), the product of *MLL* rearrangements in leukemia, also bind to telomeric chromatin. However, an effect on telomere transcription in *MLL*-rearranged (*MLL*-r) leukemia has not yet been evaluated. Here, we show increased UUAGGG repeat-containing RNA levels in *MLL*-r acute lymphoblastic leukemia (ALL) when compared to non-*MLL*-r ALL and myeloid leukemia. *MLL* rearrangements do not affect telomere length and UUAGGG repeat-containing RNA levels correlate with mean telomere length and reflect increased levels of TERRA. Furthermore, high levels of TERRA in *MLL*-r ALL occur in the presence of telomerase activity and are independent of ploidy, an underestimated source of variation on the overall transcriptome size in a cell. This *MLL* rearrangement-dependent and lymphoid lineage-associated increase in levels of TERRA supports a sustained telomere transcription by MLL-FPs that correlates with marked genomic stability previously reported in pediatric *MLL*-r ALL.

## 1. Introduction

TERRA consists of RNA polymerase II (RNA pol II)-transcribed RNA molecules heterogeneous in length which contain telomeric UUAGGG repeats at their 3′ ends and arise from transcription start sites in subtelomeric regions [1,2]. This evolutionary conserved long non-coding RNA binds to telomeres where, in addition to a telomere-length dependent regulation of telomerase activity [2,3,4,5], it has been seen to counteract the DNA-damage response to telomere shortening in rare budding yeast cells surviving telomerase deletion through the formation of TERRA RNA-telomeric DNA hybrids (R-loop) involved in homologous recombination (HR) and responsible for the maintenance of telomere length, chromosomal stability, and delay in the onset of cellular senescence [6,7].

Published results indicate different genomic origins of TERRA in mammalian cells [8,9,10,11], from only a few to several subtelomeric regions and from CpG-island and non-CpG-island-containing promoters [12,13]. Regardless of their transcriptional origin, early and current data seem to agree on the association of TERRA to nearly all chromosome ends [1,2,8,10], and now to additional non-telomeric chromosomal regions [14]. Consequently, the number of complete chromosomal sets in a cell, i.e., ploidy, which is an underestimated and non-linearly correlated source of variation in overall transcriptome size [15,16], also emerges as a potential source of variation in total levels of TERRA, both in terms of numbers of subtelomeric transcription start sites and of telomeres to which TERRA binds.

TERRA has also been shown to bind to heterogeneous nuclear ribonucleoprotein A1 (hnRNPA1) and to regulate its ability to displace replication protein A (RPA) from single-stranded telomeric DNA in favor of the binding of protection of telomeres 1 (POT1) shelterin protein and the recruitment of telomerase activity to telomeric 3′ overhangs [4,17,18]. In the telomerase-independent alternative lengthening of telomeres (ALT) cancer cells carrying a mutation in the alpha-thalassemia/mental retardation syndrome X-linked (*ATRX*) gene, RPA displacement by hnRNPA1 seems to be affected by the compromised cell-cycle regulation of TERRA [19]. Sustained levels of TERRA and sequestered hnRNPA1 around telomeres during late-S to G2/M-phase progression cause the persistent association of RPA with single-stranded DNA in highly recombinogenic nucleoprotein structures at telomeres [19,20]. R-loop formation from a persistent association of TERRA during telomeric DNA replication is also believed to engage in HR-dependent telomere lengthening in ALT cancer cells, as demonstrated by R-loop resolution, HR reduction, and accelerated telomere shortening following the overexpression of the RNA-DNA hybrid-specific endonuclease RNaseH1 [21].

High levels of TERRA also increase histone 3 lysine 9 di/trimethylation (H3K9me2/3) density and heterochromatin protein 1α (HP1α) deposition to telomeric chromatin through the recruitment of Polycomb Repressive Complex 2 (PRC2) and its histone 3 lysine 27 trimethylation (H3K27me3) activity. The resulting crosstalk between H3K27 and H3K9 methylation marks, together with increased subtelomeric DNA methylation, contribute to telomere heterochromatin formation, thus silencing telomere transcription through a TERRA-dependent negative feedback loop [22,23,24]. This leads to a decrease in levels of TERRA, as was found in human fibroblasts and cancer cell lines ectopically expressing telomerase in which artificially elongated telomeres showed increased H3K9me3 density and HP1α deposition [23].

The negative feedback regulation of TERRA seems to be altered in ALT cancer cells with heterogeneous subtelomeric DNA methylation, compromised telomere heterochromatin formation, and increased levels of TERRA [25,26,27]. Similarly, subtelomeric DNA hypomethylation and reduced telomeric H3K9me3 density occur in fibroblasts derived from immunodeficiency, centromeric region instability, and facial anomalies (ICF) syndrome type I patients carrying a mutation in the DNA methyltransferase 3b (DNMT3b) gene. The result is an impaired telomere heterochromatin formation with abnormal increases in telomeric histone 3 lysine 4 dimethylation (H3K4me2), TERRA transcription, and non-recombinogenic R-loop accumulation causing accelerated telomere shortening, telomere instability, and premature senescence [28,29,30].

TERRA protection of telomeres from the persistent activation of DNA-damage response was initially suggested by the appearance of telomeric dysfunction in ALT cancer cells and in human diploid fibroblasts respectively deprived of TERRA and its transcriptional regulator mixed-lineage leukemia (MLL) [22,31,32]. The MLL protein, also known as lysine methyltransferase 2A (KMT2A), is the founding member of the Trithorax group (TrxG) of transcriptional activators involved in epigenetic maintenance of gene expression that counteract the gene repression programs established by the Polycomb group (PcG) of transcriptional repressors [33]. MLL is proteolytically cleaved into MLL^N^ and MLL^C^ fragments held together non-covalently in a KMT2A protein complex bound to hundreds of RNA pol II-poised, developmentally regulated coding and non-coding genes [34,35]. MLL^C^ association with KMT2A protein complex subunits WDR5, RbBP5, ASH2L, and DPY30 is essential for H3K4 methyltransferase activity [34], while MLL^N^ association with LEDGF/PSIP1 and menin subunits confers KMT2A protein complex targeting to the chromatin [36,37].

The MLL protein binds to telomeric chromatin, where it contributes to the maintenance of H3K4me2/3 levels and, in cooperation with the tumor suppressor protein TP53 (p53), to the transcriptional regulation of chromosome-specific TERRA in telomere length homeostasis and at dysfunctional telomeres as part of a protective mechanism from DNA-damage response [31,32,38]. Quantitative mass spectrometry analysis has now identified MLL and its interacting proteins LEDGF/PSIP1, menin, WDR5, and RbBP5 in a TERRA interactome as components of the transcriptional machinery, while LEDGF/PSIP1 and WDR5 are among the hundreds of new proteins found to be associated with mammalian telomeric chromatin [11,39,40].

Chromosomal rearrangements of the *MLL* gene are frequently identified in pediatric, adult, and therapy-related leukemia in approximately 10% of all human leukemia, with *MLL* rearrangements causing over 70% of infant (≤1 year of age) acute lymphoblastic leukemia (ALL) and 35–50% of infant acute myeloid leukemia (AML) cases [41]. As a result of *MLL* rearrangements, MLL fusion proteins (MLL-FPs) have been identified as containing the MLL N-terminus fused with a C-terminus encoded by at least 84 different translocation partner genes [42]. The dimerization of MLL N-terminal exons resulting from in-frame partial tandem duplications of 5′ *MLL* regions (*MLL*-PTDs) has also been identified in AML patients comprising 4% of the *MLL*-r leukemia [41,42].

*MLL* chromosomal translocations with *AF4* (*AFF1*), *AF9* (*MLLT3*), *ENL* (*MLLT1*), *AF10* (*MLLT10*), *ELL,* and *AF6* (*MLLT4*) partner genes are found in approximately 85% of *MLL*-r leukemia [43], where all but *AF6* encode for nuclear proteins interacting in an elongation-assisting protein (EAP) complex [44], also known as super-elongation complex or AF4, and ENL family protein complex [45,46]. The EAP complex is believed to participate in both the physiologic and oncogenic expression of MLL target genes. MLL fusions with EAP components constitutively recruit the EAP complex to RNA pol II-poised MLL target oncogenes and lineage-associated genes, causing their sustained expression and the resulting transformation of hematopoietic progenitors [44].

Conditional expression of MLL fusions with EAP components confirmed the association of MLL-FPs to the telomeric chromatin [31]. However, despite the demonstrated association of MLL proteins with telomeres, the effect of *MLL* rearrangements on telomere transcription in human acute myeloid and lymphoblastic leukemia has not yet been analyzed. To evaluate this effect, we aimed to quantify the UUAGGG repeat-containing RNA levels in *MLL*-r and non-*MLL*-r leukemia cell lines with respect to their lineage, telomere length, and ploidy as a method for measuring differences in TERRA levels.

## 2. Materials and Methods

### 2.1. Cell Cultures

Except for ALL-PO [47], all cell lines were obtained from the American Type Culture Collection (ATCC) and the German Collection of Microorganisms and Cell Cultures (DSMZ) public repositories. K-562, Nalm-1, 697, and MV4;11 cell lines were grown in IMDM medium supplemented with 10% FCS. MV4;11 cells were cultured in the presence of GM-CSF. All of the other cell lines were cultured in RPMI 1640 medium supplemented with 10 to 20% FCS (RPMI-F).

### 2.2. ChIP Analysis

Telomeric ChIP was carried out as previously described [31]. Each sample consisted of 1 × 10^6^ cells. Chromatin was immunoprecipitated using rabbit polyclonal antibodies (pAbs), anti-MLL N terminus (M382 and M612), and corresponding preimmune sera (P), and anti-TRF2 (FC08) and the corresponding preimmune sera (P2) described previously [31]. The resulting DNA was analyzed by slot-blot hybridization using a ^32^P-end-labeled oligonucleotide 5′-(TTAGGG)_4_-3′ complementary to the telomeric CCCTAA repeats (*tel*). The filter was then stripped and hybridized with ^32^P-labeled p82H DNA, a human α-satellite sequence specific for the centromere of every human chromosome (*cen*) [31], or with a mix of ^32^P-end-labeled oligonucleotides specific for H1, H2A, H2B, and H3 histone genes (*his*). Autoradiographic signals were quantified by densitometry analysis using the National Institutes of Health ImageJ software.

### 2.3. RNA Slot-Blot Analysis

Total RNA preparations were cleaned with TURBO DNA-*free*™ (Invitrogen, Waltham, MA, USA), quantified, and then loaded at 2, 1, 0.5, and 0.25 μg per slot on a GeneScreen Plus filter (PerkinElmer, Waltham, MA, USA) and UV cross-linked. The filters were hybridized overnight with a ^32^P-end-labeled 5′-(CCCTAA)_4_-3′ oligonucleotide complementary to TERRA’s UUAGGG repeats, then stripped and hybridized with a ^32^P-end-labeled 5′-GGGAACGTGAGCTGGGTTTAGACC-3′ oligonucleotide specific to 28S rDNA as a transcriptome loading control. Autoradiography signals were quantified by densitometry analysis using ImageJ software NIH ImageJ software v1.52 from the National Institute of Health, and each UUAGGG repeats signal was normalized to the corresponding 28S rDNA signal (UUAGGG/28S) as a measurement of absolute UUAGGG repeats per transcriptome in each cell line. For each cell line, UUAGGG/28S from different blots were then averaged and expressed relative to the mean UUAGGG/28S found in non-*MLL*-r leukemia.

### 2.4. Terminal Restriction Fragment (TRF) Analysis

*RNase*-treated genomic DNA was digested with 25U of each *HinfI* and *RsaI* restriction enzyme. The fragmented DNA was quantified and fractionated in 0.7% agarose gel for 700–1000 V·h. The gels were denatured/neutralized, and the DNA was transferred onto Hybond N-plus filters (Fisher Scientific, Waltham, MA, USA). The filters were UV cross-linked and hybridized overnight with a ^32^P-end-labeled 5′-(TTAGGG)_4_-3′ oligonucleotide. Autoradiography signals were quantified by densitometry analysis, and the mean telomere length, variance, and mode were calculated using Telometric software [48].

### 2.5. Telomeric-Repeat Amplification Protocol (TRAP)

Detection of telomerase activity in cell extracts was done by the PCR-based TRAP assay using the ^32^P end-labeled TS forward and the CX reverse primers previously described [49]. Telomerase extracts were allowed to extend an excess of the TS primer (0.1 μg) for 15 min at 22 °C in 50 μL of 1X buffer B (20 mM Tris-Cl, pH 8.3; 1.5 mM MgCl2; 63 mM KCl; 0.005% Tween 20; 1 mM EGTA; 0.1 mg/mL BSA) containing Taq DNA polymerase. After the telomerase extension reaction, telomerase activity was inactivated at 65 °C for 3 min and 0.1 μg of CX oligonucleotide was added to the reaction. A PCR was performed for 27 cycles as previously indicated [49]. Products were resolved in 15% polyacrylamide gels for 1800 Vh and the gels were fixed and exposed to autoradiography. HeLa cell extracts, no added extract, and RNase A treatment of extracts were used as controls.

## 3. Results

### 3.1. Telomeric Chromatin Binding of MLL Proteins and Telomerase Activity in MLL-Rearranged Acute Lymphoblastic Leukemia

Our chromatin immunoprecipitation (ChIP) analysis revealed an association of MLL proteins with telomeric and centromeric chromatin in human diploid fibroblasts, telomerase- and ALT-dependent immortalized human epithelial cells, and MLL-r and non-MLL-r leukemia for the first time [31]. MLL proteins revealed an association with telomeric and centromeric chromatin of MLL-r pre-B-ALL (ALL-PO and RS4;11) and non-MLL-r pre-B-ALL (697 and Nalm-6), T-ALL (Jurkat), and AML (U-937) cell lines (Table 1 and Figure 1A). Similar to endogenous MLL proteins, ectopic MLL-AF4, and MLL-AF9 fusion proteins have also shown an association with telomeric chromatin through their MLL N-terminus (MLL3AT and MLLDT) when conditionally expressed in non-MLL-r U-937 myeloid leukemia stable transfectants (Figure 1B,C). As with the non-MLL-r U-937 AML cell line, the telomeric-repeat amplification protocol (TRAP) detected telomerase activity in ALL-PO and RS4;11 cell lines as indicative of a telomerase-based mechanism of telomere maintenance associated with MLL-r ALL (Figure 1D).

### 3.2. Increased Levels of UUAGGG Repeat-Containing RNA in MLL-r ALL

To analyze the effect of MLL rearrangements expression on telomere transcription, we quantified the levels of UUAGGG repeat-containing RNA in a panel of MLL-r and non-MLL-r myeloid and lymphoblastic leukemia cell lines (Table 1) by slot-blot analysis with a 5′-(CCCTAA)_4_-3′ probe, which gives an unbiased quantification of UUAGGG repeat content in a cell (Figure 2A). For each cell line, UUAGGG repeats signals from different blots were normalized to the corresponding 28S rRNA oligonucleotide-specific signals as an estimate of UUAGGG repeats content per transcriptome (UUAGGG/28S), averaged and expressed relative to the transcriptome-normalized UUAGGG repeats mean level in non-MLL-r leukemia cell lines. We used a two-tailed *t*-test and two-way analysis of variance (ANOVA) to assess the effects of MLL rearrangement expression and cell lineage on the transcription of TERRA (Figure 2B).

Overall, we did not find significant variation in UUAGGG repeats content per transcriptome between MLL-r and non-MLL-r leukemia (*p* = 0.164) or between myeloid (light columns) and lymphoid (dark columns) cell lineage (*p* = 0.082). However, Fisher’s least significant difference (Fisher’s LSD) analysis showed significant differences in UUAGGG/28S levels between MLL-r and non-MLL-r ALL (*p* < 0.05) and between MLL-r ALL and MLL-r AML (*p* < 0.02), with 2.3- to 2.6-fold higher levels of UUAGGG repeat-containing RNA found in MLL-r ALL, respectively (Figure 2B). We found no significant difference in UUAGGG/28S levels between MLL-r and non-MLL-r myeloid leukemia (*p* = 0.814) or between non-MLL-r myeloid and lymphoid cell lineage (*p* = 0.944).

### 3.3. MLL Rearrangements Do Not Affect Telomere Length in Leukemia

Levels of UUAGGG repeat-containing RNA per cell, both total and chromatin-bound, have been directly correlated with telomere length and telomeric CCCTAA template repeats content [23,25,28]. To determine whether a telomere length effect may account for differences in UUAGGG repeats content per transcriptome seen between MLL-r and non-MLL-r ALL or between MLL-r ALL and MLL-r AML, we assessed the mean telomere length (MTL) of each cell line by HinfI/RsaI terminal restriction fragment (TRF) analysis (Figure 3A). A two-way ANOVA analysis indicated no significant difference in MTL between MLL-r and non-MLL-r leukemia (*p* = 0.157), regardless of cell lineage (myeloid, *p* = 0.197; lymphoid, *p* = 0.470) (Figure 3B). This excludes MLL rearrangements as a significant cause of variation in telomere length and therefore in telomeric CCCTAA template repeats content in both myeloid and lymphoid leukemia. However, the U-937, REH, and Karpas-45 cell lines showed an MTL greater than 10kb, well above the panel’s average of 6.2 kb. This may result in a marked telomere length effect on TERRA levels in U-937, REH, and Karpas-45 when compared to leukemia cell lines with lower MTL.

### 3.4. UUAGGG Repeat-Containing RNA Levels in MLL-r ALL Correlate with Telomere Length and Reflect Increased Levels of TERRA

We then evaluated whether a correlation exists between transcriptome normalized UUAGGG repeats content (UUAGGG/28S) and MTL in each leukemia lineage and MLL genotype analyzed (Figure 4A). A linear correlation analysis revealed a significant positive correlation between UUAGGG/28S levels and MTL in MLL-r leukemia (R = 0.82; *p* < 0.002), which includes a strong positive correlation identified in MLL-r ALL (R = 0.92; *p* < 0.03) (Figure 4B). We also found a moderate positive correlation between UUAGGG/28S levels and MTL in both non-MLL-r (R = 0.43; *p* = 0.39) and MLL-r (R = 0.44; *p* = 0.38) myeloid leukemia, as well as in all lymphoid leukemia cell lines analyzed (R = 0.46; *p* = 0.18) although the Pearson’s correlation coefficient was not statistically significant in any of these correlation analyses (Figure 4B).

With the level of UUAGGG repeats-containing RNA per cell correlated with the length of the telomeric DNA template [23,25,28], an estimate of TERRA levels per transcriptome (TERRA/28S) can be calculated by expressing the UUAGGG repeats content per transcriptome relative to MTL [(UUAGGG/28S)/MTL]. When the UUAGGG/28S level of each cell line was divided by the corresponding MTL, we found transcriptome-normalized TERRA levels 1.8-fold higher in MLL-r than in non-MLL-r leukemia (*p* < 0.01) but not significantly different between lymphoid and myeloid leukemia (*p* = 0.098). MLL-r ALL showed TERRA/28S levels 2.8-fold higher than in non-MLL-r ALL (*p* < 0.001) and 1.9-fold higher than in MLL-r AML (*p* < 0.005) (Figure 4C). Among the MLL-r ALL cell lines analyzed, Karpas-45 retained high levels of TERRA despite a greater than average MTL and marked telomere length effect. We did not find a significant difference in TERRA levels per transcriptome in MLL-r compared to non-MLL-r myeloid leukemia (*p* = 0.678) or between non-MLL-r myeloid and lymphoid cell lineage (*p* = 0.512) (Figure 4C). Overall, UUAGGG repeat-containing RNA levels in MLL-r ALL positively correlate with MTL and reflect increased levels of TERRA.

### 3.5. Increased Levels of TERRA in MLL-r ALL Are Independent of Ploidy

At present, no data is available on potential changes in TERRA levels per cell related to ploidy, hereafter referred to as TERRA dosage response. For instance, the effects of whole-genome duplication, also known as tetraploidy, on TERRA levels can range from dosage compensation, with no change in TERRA levels relative to diploid cells; to a 1:1 dosage effect, with a doubling of TERRA transcripts per doubling of telomeres; or any intermediate response [15]. In tetraploid cells with dosage compensation of rRNA, which accounts for approximately 80% of the overall transcriptome size, TERRA levels per transcriptome (TERRA/28S) directly correlate with TERRA levels per cell (TERRA/cell) (Figure 5A). TERRA levels per telomere (TERRA/telomere), as a measure of TERRA dosage response, can be inferred from TERRA levels per transcriptome when divided by ploidy [(TERRA/28S)/ploidy]. On the other hand, assuming a 1:1 rRNA dosage effect, TERRA/telomere directly correlates with TERRA/28S levels (Figure 5A).

Of all the leukemia cell lines analyzed, thirteen out of twenty-three carry a hypo-, hyper-, or near diploid karyotype, thus minimizing a TERRA dosage response in five myeloid and eight lymphoid leukemia, including four out of five MLL-r ALL. In the remaining ten cell lines, three non-MLL-r myeloid leukemia carry a hypo- to hyper-triploid karyotype while four MLL-r AML, two non-MLL-r ALL, and one MLL-r ALL present a hypo- to near tetraploid karyotype, all with TERRA dosage response from changes in total telomere number (Table 1 and Figure 5B).

Assuming rRNA dosage compensation in all polyploid leukemia cell lines analyzed, transcriptome-normalized TERRA levels divided by ploidy [(UUAGGG/28S/MTL)/ploidy] confirmed significantly higher TERRA levels per telomere in MLL-r ALL than non-MLL-r ALL (3.2-fold; *p* < 0.0001) or MLL-r AML (2.3-fold; *p* < 0.0005) (Figure 5C). On the other hand, assuming a 1:1 rRNA dosage effect, the TERRA/telomere directly correlates with TERRA levels per transcriptome (UUAGGG/28S/MTL) in each cell line (see Figure 4C). Therefore, the TERRA dosage response exerts a minimal effect on the increased levels of TERRA in MLL-r ALL. Overall, increased levels of TERRA in MLL-r ALL are independent of ploidy, an underestimated source of potential variation in total levels of TERRA in a cell.

## 4. Discussion

The chromatin modifier and transcriptional regulator MLL binds to telomeres and regulates TERRA transcription [31]. Our ChIP analysis has revealed an association of MLL proteins with telomeric chromatin in both non-*MLL*-r and *MLL*-r leukemia, with ectopically expressed MLL-AF4, MLL-AF9, and MLL^N^ deletion mutants displaying the binding of MLL-FPs to telomeres through their MLL N-terminus, in support of potential deregulated telomere transcription in *MLL*-r leukemia.

Here, we report significantly higher levels of TERRA in *MLL*-r than non-*MLL*-r leukemia, an *MLL* rearrangement-dependent effect that associates with lymphoid lineage. Increased levels of TERRA in *MLL*-r ALL are independent of telomere length and ploidy, which can affect the telomeric CCCTAA template repeats content and the number of subtelomeric TERRA transcription start sites, respectively, and the telomeres to which TERRA binds. A comparison of four near diploid *MLL*-r pre-B-ALL and one hypotetraploid *MLL*-r T-ALL with four near diploid non-*MLL*-r pre-B-ALL and one hypotetraploid Burkitt lymphoma accounts for the most significant difference between TERRA levels. The telomerase activity detected in ALL-PO and RS4;11 cell lines together with the lack of exceedingly long and heterogeneous ALT telomeres support increased levels of TERRA in *MLL*-r ALL in the presence of telomerase-dependent telomere elongation. The increased levels of TERRA found in *MLL*-r ALL contrast with their reduction during the progression of telomerase-positive astrocytoma or squamous cell, hepatocellular, and endometrial carcinoma [51,52,53,54], suggesting an advantage conferred by high levels of TERRA to this poor-prognosis leukemia subtype.

MLL expression and TERRA transcription are regulated during the cell cycle, with MLL peaking at G1/S-phase transition and both MLL and TERRA levels decreasing in the S-phase of DNA replication [55,56]. In late-S to G2/M-phase progression, when TERRA reaches the lowest levels, MLL increases again in the G2-phase, reaching a second peak at G2/M-phase transition. MLL remains associated with chromosomes throughout the M-phase, where it functions as a mitotic bookmark to facilitate post-mitotic transcriptional reactivation of target genes, and then decreases in the early G1-phase [33,57]. Interestingly, the highest levels of TERRA have been observed in the early G1-phase [56], as expected from MLL target genes bookmarked for rapid transcriptional reactivation following mitotic exit. In *MLL*-r ALL, the MLL-FP levels remain constant throughout the cell cycle due to their impaired interaction with Skp2 and Cdc20, two specialized E3 ligases responsible for the cell cycle-regulated degradation of MLL by the ubiquitin/proteasome system [55]. A refractory MLL-FP degradation in *MLL*-r ALL may therefore lead to sustained telomere transcription, and the analysis of synchronized *MLL*-r ALL cells will be required to confirm elevated levels of TERRA throughout the cell cycle.

Thus, as with *ATRX* mutation in ALT cancer cells and *DNMT3B* mutation in ICF syndrome type I fibroblasts, *MLL* rearrangements in *MLL-r* ALL would also appear to prevent a negative feedback regulation of TERRA. The analysis of subtelomeric DNA methylation and telomeric H3K4 and H3K27 methylation density will be necessary to confirm an MLL-FP sustained transcriptional activity that contrasts the PRC2 silencing and heterochromatin formation activities at telomeres [58]. Furthermore, the association of WDR5, RbBP5, LEDGF, and menin to telomeres as revealed by several quantitative mass spectrometry analyses of the TERRA interactome [11,39,40], as well as the association of MLL to telomeric chromatin in both telomerase- and ALT-dependent cells [31], suggest an in-depth analysis of the role of the KMT2A complex and efficacy of its inhibitors in the regulation of telomere transcription into TERRA.

It is known that TERRA and telomere transcription do not affect telomerase-dependent telomere elongation in human cancer cells [3]. ATRX depletion in telomerase-positive HeLa cells has shown the sustained transcription of TERRA during late-S to G2/M-phase progression without causing telomerase inactivation or affecting telomere length [19]. Similarly, a sustained transcription of TERRA in *MLL*-r ALL occurs in the presence of telomerase activity without affecting telomere length. On the other hand, the ectopic expression of telomerase counteracts telomere instability caused by R-loop-accelerated telomere shortening in ICF syndrome type I fibroblasts with elevated levels of TERRA [30,59]. The same occurs during ICF syndrome type I fibroblast reprogramming, where the abnormal telomere phenotype is overridden and senescence is suppressed by telomerase-dependent telomere elongation despite persistent subtelomeric DNA hypomethylation and high levels of TERRA. Therefore, in the presence of telomerase activity, the loss of the negative feedback regulation of TERRA does not alter telomere stability, which is instead preserved from the instability caused by R-loop-accelerated telomere shortening.

Importantly, the sustained transcription of TERRA in *MLL*-r ALL does not appear to adversely affect genome stability, as indicated by lower rates of somatic alterations including copy number abnormalities such as deletions and amplifications, single-nucleotide mutations, or structural variations found in pediatric *MLL*-r ALL versus non-*MLL*-r ALL or other pediatric and adult cancers [60,61,62]. Recent studies on telomerase-positive cancer cells reveal a primary *cis* function of TERRA in facilitating the completion of telomeric DNA replication and, consequently, in maintaining telomere and genome stability [63]. Besides a *cis*-protective function at the telomeres, TERRA also appears to bind to extratelomeric chromatin in mouse embryonic stem cells, exerting *trans-*regulatory activities at nearby genes [14]. In conclusion, the sustained transcription of TERRA may provide a yet-to-be-identified telomeric and/or extratelomeric advantage to *MLL*-r ALL as part of lymphoid transcriptional reprogramming associated with *MLL* rearrangements. It remains to be determined whether this advantage translates into greater genomic stability, which could be therapeutically exploited by targeting TERRA to reverse it.

## Figures and Tables

**Figure 1 biomedicines-11-00925-f001:**
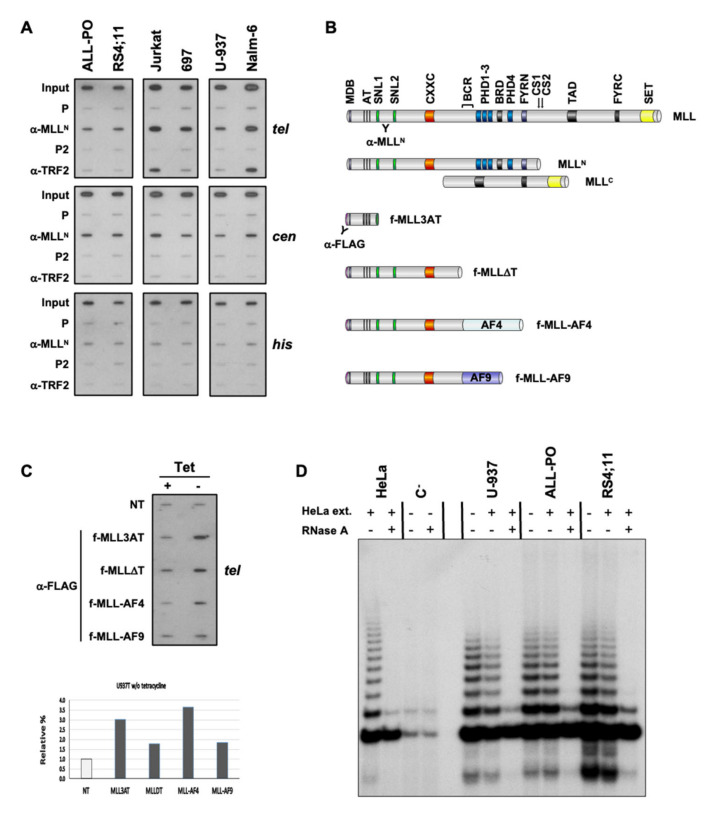
Telomeric chromatin binding of MLL proteins and telomerase activity in *MLL*-r ALL. (**A**) Representative ChIP analysis of *MLL*-r pre-B-ALL (ALL-PO and RS4;11) and non-*MLL*-r pre-B-ALL (Nalm-6 and 697), T-ALL (Jurkat), and AML (U-937) cell lines. Chromatin was immunoprecipitated using polyclonal antibodies against MLL^N^ (see MLL proteins schematic on panel B for epitope localization) and TRF2 proteins, blotted and hybridized sequentially with ^32^P-end-labeled oligonucleotides specific for telomeric DNA repeats (*tel*) and histones coding sequences (*his*), and a human α−satellite DNA sequence specific for the centromere of every human chromosome (*cen*). The input signal consists of 2% of the amount of chromatin used with each immunoprecipitation. P and P2 are pre-bleed sera, respectively, of α-MLL^N^ and α-TRF2 pAbs. (**B**) Schematic of endogenous and ectopic MLL proteins analyzed by telomeric ChIP. MDB: Menin binding domain; AT: AT hook motifs; SNL1/2: Speckled nuclear localization domains; CXXC: CXXC zinc finger domain; BCR: Breakpoint cluster region; PHD1-4: Plant homeodomain fingers; BRD: Bromodomain; FYRN/C: “FY-rich” domain N- and C-terminal; CS1/2: Taspase 1 endopeptidase cleavage sites; TAD: Transactivation domain; SET: Su(var)3-9, Enhancer-of-zeste, and Trithorax domain. (**C**) Empty vector-transfected (NT) U-937 cells and clones conditionally expressing FLAG-tagged MLL-AF4 and MLL-AF9 fusion proteins and N-terminal fragments retaining 410 (MLL3AT) and 1436 (MLLΔT) amino acids of MLL^N^ under tetracycline (*Tet*)-off conditions were grown in the presence (+) or absence (–) of 1 μg/mL *Tet.* An equal number of cells were then analyzed by ChIP for conditionally expressed MLL protein binding to telomeric DNA repeats (*tel*) using α-FLAG pAbs. We quantified MLL protein signals by ImageJ densitometry analysis, normalized each (–) *Tet* signal towards the corresponding (+) *Tet* signal (dark columns), and expressed them relative to the NT signal (light column). (**D**) Telomere repeats amplification protocol (TRAP) analysis of non-*MLL*-r U-937, and *MLL*-r ALL-PO and RS4;11 cell lines. All three cell lines revealed telomerase activity for telomere length maintenance. HeLa cell extract and treatment with RNase were used as control.

**Figure 2 biomedicines-11-00925-f002:**
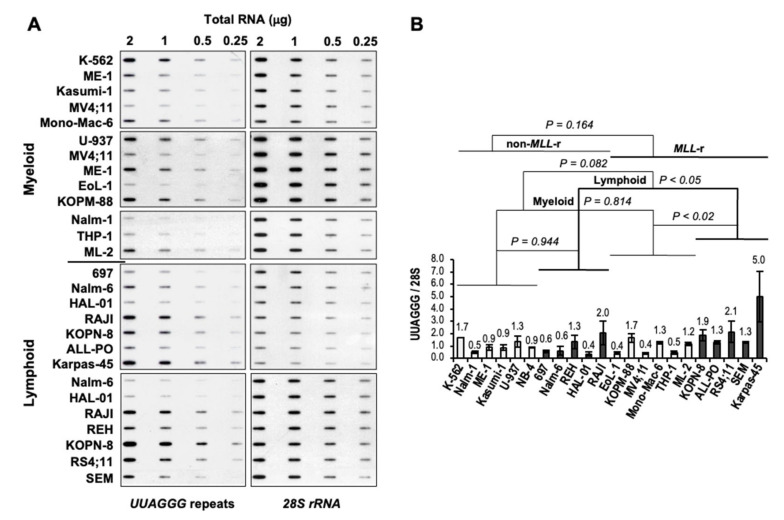
Increased levels of UUAGGG repeat-containing RNA in *MLL-r* ALL. (**A**) Representative TERRA slot-blot analysis of total RNA extracted from leukemia cell lines. *DNase*-treated total RNA was loaded at 2.0, 1.0, 0.5, and 0.25 μg per slot on a filter and UV cross-linked. Filters were sequentially hybridized with ^32^P-end-labeled oligonucleotides specific for TERRA [5′-(CCCTAA)_4_-3′], and 28S rRNA (5′-GGGAACGTGAGCTGGGTTTAGACC-3′) as a transcriptome loading control. We quantified autoradiography signals by ImageJ densitometry analysis. (**B**) Quantitation of UUAGGG repeat-containing RNA levels in myeloid (light columns) and lymphoblastic (dark columns) leukemia cell lines carrying (or not) *MLL* rearrangements. For each cell line, UUAGGG repeats signals from different blots were normalized to the corresponding 28S rRNA signals for equal transcriptome loading (UUAGGG/28S), averaged, and expressed relative to the mean value of UUAGGG/28S levels in non-*MLL*-r leukemia cell lines. Error bars represent the standard error of the mean (s.e.m.) between two and five separate experiments. We calculated the *p*-value through a two-way ANOVA analysis.

**Figure 3 biomedicines-11-00925-f003:**
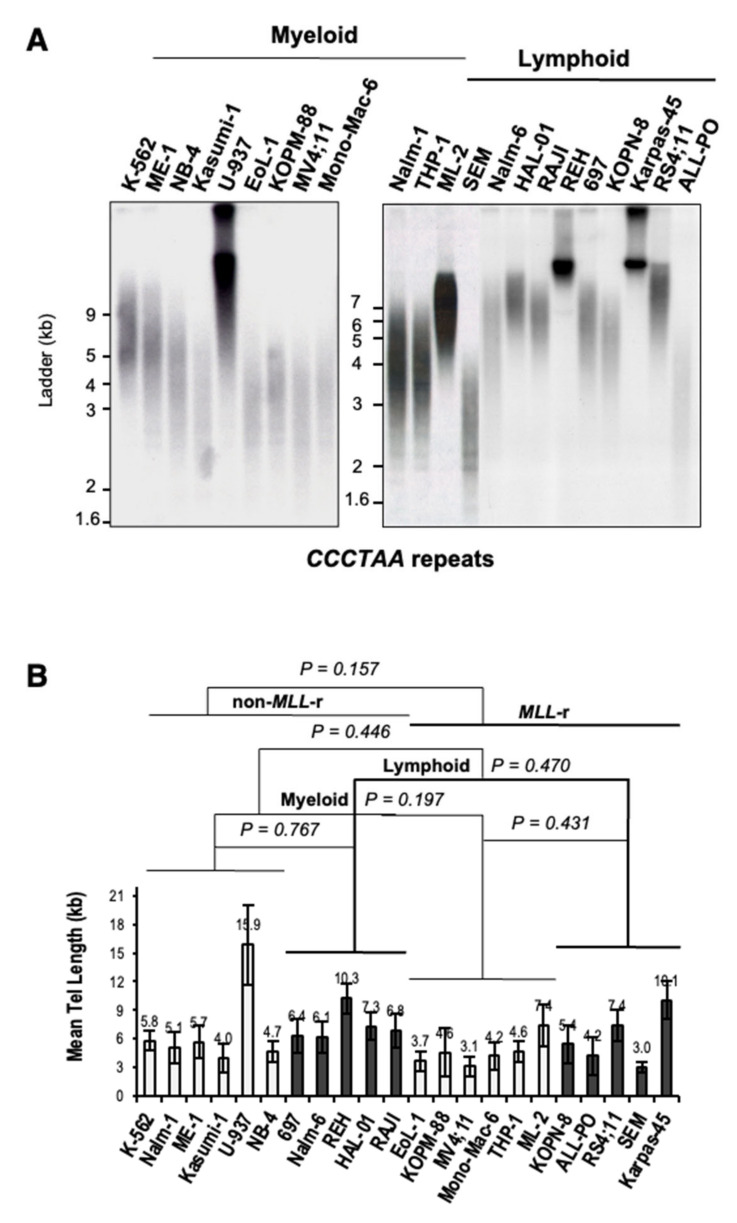
*MLL* rearrangements do not affect telomere length in leukemia. (**A**) Representative TRF analysis of myeloid and lymphoblastic leukemia cell lines carrying (or not) *MLL* rearrangements. Filters were hybridized with ^32^P-end-labeled 5′-(TTAGGG)_4_-3′ oligonucleotide complementary to the telomeric leading-strand CCCTAA repeats. (**B**) Telomere length analysis of myeloid and lymphoblastic leukemia cell lines. The duplex-repeat signal intensity was measured by densitometry analysis, and the mean telomere length (MTL) was calculated using Telometric software (see Materials and Methods). MTL value (kb) and standard deviation bars are indicated. We calculated the *p*-value by two-way ANOVA analysis.

**Figure 4 biomedicines-11-00925-f004:**
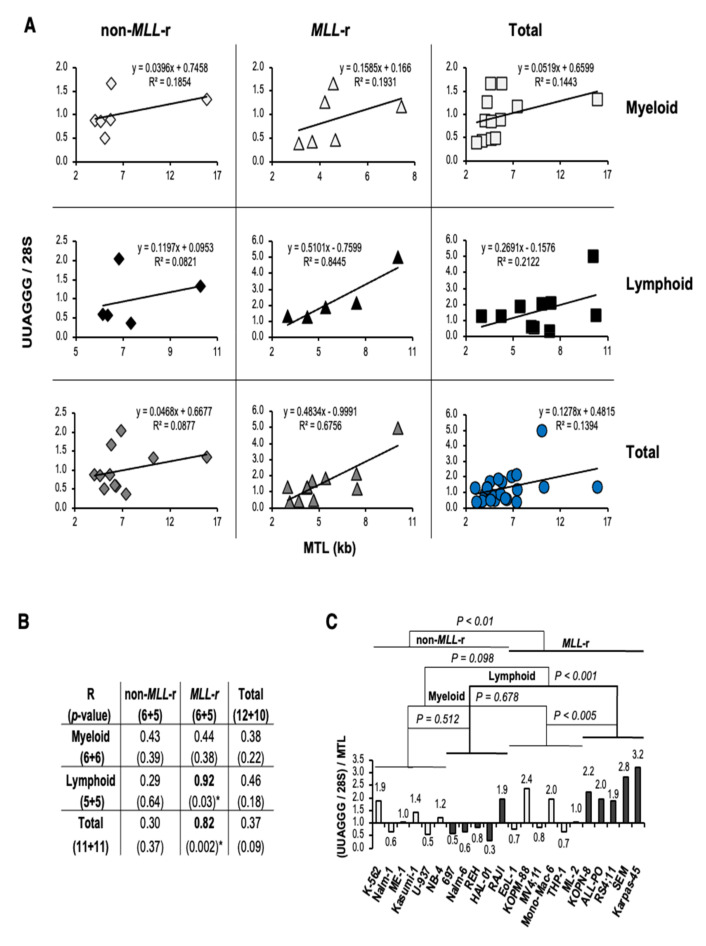
UUAGGG repeat-containing RNA levels in *MLL-r* ALL correlate with telomere length and reflect increased levels of TERRA. (**A**) Linear correlation analysis between UUAGGG repeat-containing RNA level per transcriptome (UUAGGG/28S) and mean telomere length (MTL) in leukemia cell lines of myeloid and lymphoid lineage carrying (*MLL*-r) or not (non-*MLL*-r) *MLL* rearrangements. Best trendline, linear trendline equation, and squared Pearson’s correlation coefficient (R^2^) are indicated. (**B**) Pearson’s correlation coefficient (R) and relative two-tailed probability value (*p*) from the linear correlation analysis. Statistically significant Pearson’s correlation coefficients are indicated in bold (*). (**C**) An estimate of transcriptome-normalized TERRA levels (TERRA/28S) was calculated by expressing the UUAGGG repeat-containing RNA level per transcriptome (UUAGGG/28S) relative to MTL. The TERRA/28S level [(UUAGGG/28S)/MTL)] of each cell line was expressed relative to the mean TERRA/28S value in non-*MLL*-r leukemia. We performed statistical analysis by two-way ANOVA.

**Figure 5 biomedicines-11-00925-f005:**
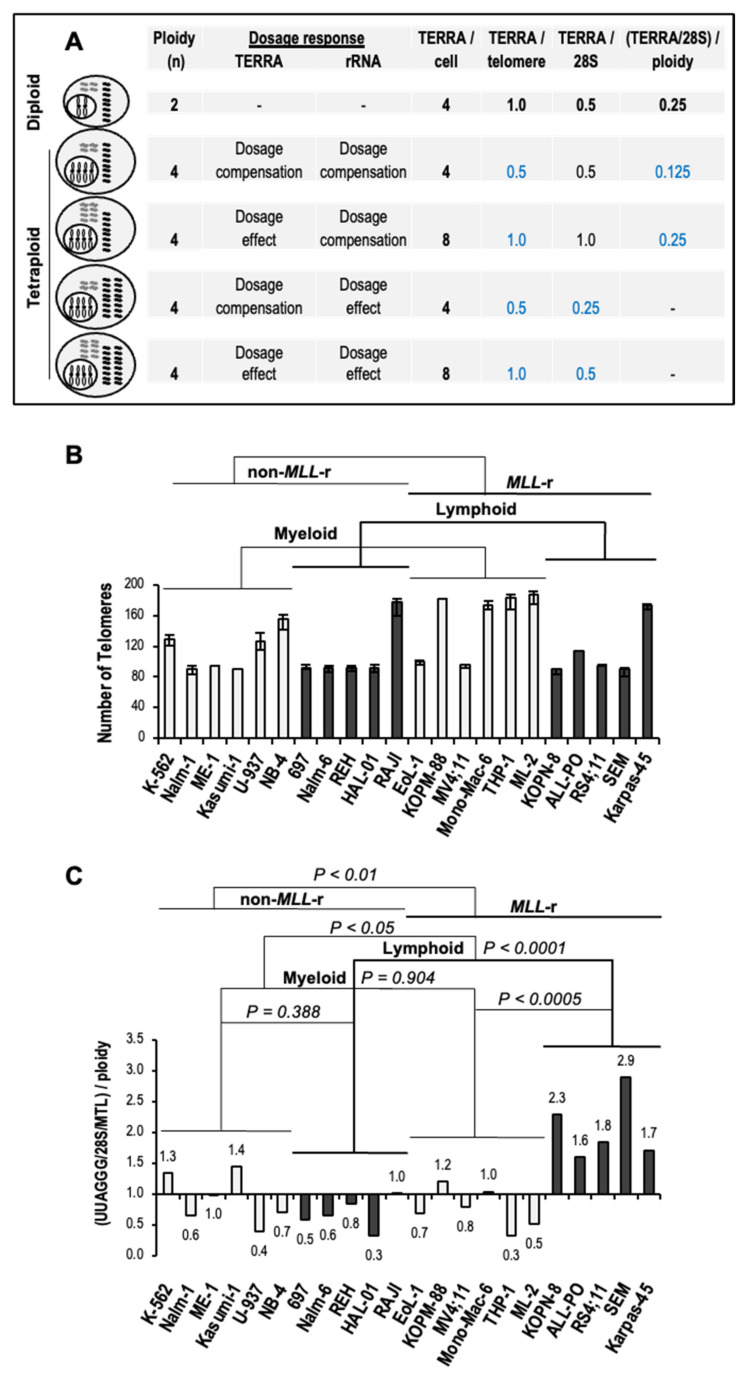
Increased TERRA levels in *MLL-r* ALL are independent of ploidy. (**A**) Potential TERRA dosage response in tetraploidy. The effects of chromosome set quadrupling (gray rods in the nucleus) on telomere transcription relative to diploid cells can go from dosage compensation, with no change in TERRA levels (gray wavy lines); to a 1:1 dosage effect, with a doubling in TERRA levels per doubling of telomeres; or intermediate responses (not illustrated). The same goes for rRNA (black wavy lines). In tetraploid cells with rRNA dosage compensation, TERRA levels per cell (TERRA/cell) correlate well with transcriptome-normalized TERRA levels (TERRA/28S) when compared to diploid cells, regardless of TERRA dosage response. TERRA levels per telomere (TERRA/telomere) are inferred from TERRA/28S when divided by ploidy. In tetraploid cells with a 1:1 rRNA dosage effect, TERRA/28S is directly correlated with the TERRA/telomere. (**B**) The total number of telomeres in each analyzed myeloid (light columns) and lymphoblastic (dark columns) leukemia cell line carrying (or not) *MLL* rearrangements, as obtained from the cytogenetic analysis of the German Collection of Microorganisms and Cell Cultures (DSMZ) public repository (see Table 1). Bars indicate the range of chromosome ends number. (**C**) TERRA/telomere levels as a measure of TERRA dosage response by polyploid cells with rRNA dosage compensation. Slot-blot UUAGGG repeat signals were measured by densitometry, normalized to the corresponding 28S rRNA signals, averaged, and then corrected for MTL (UUAGGG/28S/MTL). The obtained transcriptome-normalized TERRA levels were then divided by ploidy [(UUAGGG/28S/MTL)/ploidy] as a measure of TERRA/telomere levels and expressed relative to the average in non-*MLL*-r leukemia cell lines. We performed statistical analysis by two-way ANOVA.

**Table 1 biomedicines-11-00925-t001:** *MLL*-rearranged and non-*MLL*-rearranged leukemia cell lines **^a^** Cell lines obtained from the German Collection of Microorganisms and Cell Cultures (DSMZ) public repository (see https://www.dsmz.de, accessed on 16 October 2022). **^b^** Clinical data as reported at the DSMZ and in the literature: type of disease, age (years)/sex, source of material from which cell line was established [50]. **^c^** Cytogenetic analysis at the DSMZ confirmed the chromosomal abnormality. **^d^** RT-PCR at the DSMZ showed the expression of the fusion gene. (*) Karyotype provided by Giovanni Giudici (Centro Ricerca Tettamanti, Clinica Pediatrica Università Milano-Bicocca, Ospedale S. Gerardo, Monza, Italy). Abbreviations: ALL, acute lymphoblastic leukemia; AML, acute myeloid leukemia; APL, acute promyelocytic leukemia; CML chronic myeloid leukemia; pre-B, B-cell precursor; T-ALL, T-cell ALL; BM, bone marrow; PB, peripheral blood; PE, pleural effusion; LM, left maxilla; F, female; M, male; PTD, partial tandem duplication. **^e^** The ploidy level is calculated as the total number of chromosomes divided by 23.

	Cell Type	Cell Line	Clinical Data ^b^	Cytogenetic	Fusion Gene	Karyotype	Ploidy ^e^
	**non-*MLL*-rearranged leukemia**						
1	Chronic myeloid leukemia	K-562	CML, 53 F, PE	t(9;22)(q34;q11)	*BCR-ABL* **^d^**	Hypotriploid—64(61–68)<3n>	2.8
2	Chronic myeloid leukemia	NALM-1 **^a^**	CML, 3 F, PB	t(9;22)(q34;q12) **^c^**	*BCR-ABL*	hypodiploid with 5% polyploidy—45(42–47)<2n>	2.0
3	Acute myeloid leukemia	ME-1 **^a^**	AML, M4eo, 40 M, PB	inv(16)(p13;q22)	*CBFb-MYH11* **^d^**	near diploid—47<2n>	2.0
4	Acute myeloid leukemia	KASUMI-1 **^a^**	AML, M2, 7 M, PB	t(8;21)(q22;q22) **^c^**	*AML1-ETO* **^d^**	Hypodiploid—45<2n>	2.0
5	Acute myeloid leukemia	U-937	AML, M5, 37 M, PE	t(10;11)(p13;q14)	*CALM-AF10*	Hypotriploid—63(58–69)<3n>	2.7
6	APL, Acute promyelocytic leukemia	NB-4 **^a^**	AML, M3, 23 F, BM	t(15;17)(q22;q11-12) **^c^**	*PML-RARa* **^d^**	hypertriploid with 3% polyploidy—78(71–81)<3n>	3.4
7	B cell precursor leukemia	697 **^a^**	pre-B ALL, 12 M, BM	t(1:19)(q23;p13) **^c^**	*E2A-PBX* **^d^**	near diploid—46(45–48)<2n>	2.0
8	B cell precursor leukemia	NALM-6 **^a^**	pre-B ALL, 19 M, PB	t(5;12)(q33;p13) **^c^**	*TEL-PDGFRb*	near diploid—46(4–47)<2n>	2.0
9	B cell precursor leukemia	REH **^a^**	pre-B ALL, 15 F, PB	t(12;21)(p13;q22) **^c^**	*TEL-AML1* **^d^**	Pseudodiploid—46(44–47)<2n>	2.0
10	B cell precursor leukemia	HAL-01 **^a^**	pre-B ALL, 17 F, PB	t(17;19)(q22;p13) **^c^**	*E2A-HLF* **^d^**	near diploid with 4% polyploidy; 46(43–48)<2n>	2.0
11	Burkitt lymphoma	RAJI **^a^**	Burkitt Lymphoma, 12 M, LM	t(8;14)(q24;q32) **^c^**	*IgH-cMYC*	hypotetraploid with 12% polyploidy—89(80–91)<4n>	3.9
12	T cell leukemia	JURKAT **^a^**	T-ALL, 14 M, PB	add(2)(p21)/del(2)(p23)x2	-	hypotetraploid karyotype with 7.8% polyploidy—87(78–91)<4n>	3.8
	***MLL*-** **rearranged leukemia**						
13	Acute myeloid (eosinophilic) leukemia	EOL-1 **^a^**	AML, 33 M, PB	del(4)(q12)x2 **^c^**	*FIP1L1-PDGFRa* & *MLL*-PTD	hyperdiploid with 7.5% polyploidy—50(48–51)<2n>	2.2
14	Acute monocytic leukemia	KOPM-88	AML, 11 M, PB	*MLL*-PTD	*MLL*-PTD	near tetraploid—91<4n>	4.0
15	Acute monocytic leukemia	MV4;11 **^a^**	AML, M5, 10 M, PB	t(4;11)(q21;q23) **^c^**	*MLL-AF4* **^d^**	Hyperdiploid—48(46–48)<2n>	2.1
16	Acute monocytic leukemia	MONO-MAC-6 **^a^**	AML, M5, 64 M, PB	t(9;11)(p22;q23) **^c^**	*MLL-AF9*	hypotetraploid with near-diploid (8%) and polyploid (17%) sidelines—84–90<4n>	3.8
17	Acute monocytic leukemia	THP-1 **^a^**	AML, 1 M, PB	t(9;11)(p22;q23) **^c^**	*MLL-AF9*	near-tetraploid—94(88–96)<4n>	4.1
18	Acute myelomonocytic leukemia	ML-2 **^a^**	AML, M4, 26 M, PB	t(6;11)(q27;q23) **^c^**	*MLL-AF6* **^d^**	near tetraploid—92(84–94)<4n>	4.0
19	B cell precursor leukemia	KOPN-8 **^a^**	pre-B ALL, <1 F, PB	t(11;19)(q23;p13) **^c^**	*MLL-ENL* **^d^**	hypodiploid with 4% polyploidy—45(42–45)<2n>	2.0
20	B cell precursor leukemia	ALL-PO	pre-B ALL, <1 F, BM	t(4;11)(q21;q23)	*MLL-AF4*	hyperdiploid with 20% polyploidy—57<2n> ^(^*^)^	2.5
21	B cell precursor leukemia	RS4;11	pre-B ALL, 32 F, BM	t(4;11)(q21;q23) **^c^**	*MLL-AF4* **^d^**	Hyperdiploid—47/48<2n>	2.0
22	B cell precursor leukemia	SEM **^a^**	pre-B ALL, 5 F, PB	t(4;11)(q21;q23) **^c^**	*MLL-AF4* **^d^**	hypodiploid with 1.5% polyploidy—45(40–46)<2n>	2.0
23	T cell leukemia	KARPAS-45 **^a^**	T-ALL, 2 M, BM	t(X;11)(q13;q23) **^c^**	*MLL-AFX1*	hypotetraploid with 8% polyploidy—87(84–88)<4n>	3.8

## Data Availability

Not applicable.

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
