# Peer review of "Telomere Transcription in MLL-Rearranged Leukemia Cell Lines: Increased Levels of TERRA Associate with Lymphoid Lineage and Are Independent of Telomere Length and Ploidy"

_biomedicines, 2023, doi:10.3390/biomedicines11030925_

Round 1

Reviewer 1 Report

MLL (KMT2A ) wild and rearranged bind to telemeres and regulate TERRA transcription. The influence on telomere transcription of TERRA wasn't up to now examined in MLL rearranged leukemias. The authors examined that problem on leukemic cell lines myeloid and lymphoid with MLL rearranged and not ones. The conclusions drawn from the work are interesting and further contribute to our knowledge of MLL leukemias. The work was well planned and carried out and conclusions are clear.

I do not have any remarks.

Additional Comments:

I think the topic is original and relevant and fills  up the gap of the knowledge but I would change the title which doesn’t say about acute leukemia in patients but in leukemic cell lines.
I would propose as follows: „Telomere transcription in MLL-rearranged leukemic cel lines ….” .

The methodology is sufficient to draw conclusions as they were.
The tables and figures are described comprehensively

Author Response

We thank the reviewer for taking the time to review our manuscript and appreciate the positive comments on our study, methodology, and conclusions.

We find the reviewer’s suggestion to emphasize our work as centered on the usage of hematopoietic cell lines quite pertinent. Accordingly, as proposed by the reviewer, we are changing the title in:

“Telomere transcription in MLL-rearranged leukemia cell lines: increased levels of TERRA associate with lymphoid lineage and are independent of telomere length and ploidy”

Sincerely,

Corrado Caslini

Reviewer 2 Report

The authors investigated the effect of MLL rearrangements on TERRA transcription, in a series of MLL rearranged and not-rearranged myeloid and lymphoid cell lines. They confirmed the association with telomeric chromatin also in MLL rearranged cell lines. However they demonstrated a significant increased transcriptome activity in MLL-r ALL compared to MLL-r AML and non MLL-r cell lines. Moreover they observe a strong positive correlation between UUAGGG/28S content and mean telomere length in MLL-r ALL and wit TERRA levels, justifying the survival advantage of this type of leukemic cells and the worse prognosis in this leukemia subtype.

The paper is very interesting, experiments well described and clear, as well as results, tables and figures and adds new important information on the biology of MLL-R ALL.

I have just few questions:

1.     How did they explained the different effect in telomere activity in MLL-r ALL and MLL-r AML with the same rearrangement partners (B precursor ALL vs monocytic AML?)

2.     Did they have any idea on TERRA modulation in ALL or AML patients?

3.     How could this information affect leukemia management? Could be hypothesized a target therapy counteracting the increased TERRA activity? Pleas add a sentence in the discussion.

Author Response

We thank the reviewer for the interest expressed toward our study and the appreciation of the clarity of the experimental procedure and description of the results, tables, and figures. We are particularly grateful for the interesting questions, which allowed us to further extend the vision of our study's possible implications on the biology of MLL-r ALL.

  1. How did they explain the different effects in telomere activity in MLL-r ALL and MLL-r AML with the same rearrangement partners (B precursor ALL vs monocytic AML?)

Our findings suggest that an increased expression of TERRA correlates with MLL translocations in ALL. This appears to be a lineage-specific MLL oncoprotein instructive effect (Cano et al., 2008), as evidenced by differences in levels of TERRA between lymphoblastic (RS4;11; SEM and ALL-PO) and myeloid (MV4;11) leukemia cell lines bearing the same t(4;11) translocation (Figure 2). The explanation for the lack of significant effects on TERRA levels in MLL-rearranged myeloid leukemia is currently unclear, and it may depend on poorly understood differences in telomere biology between lineages that could impact pathogenesis, disease progression, and even therapeutic approaches.

It is known that, when expressed in hematopoietic progenitors, MLL oncoproteins upregulate self-renewal-associated genes typically expressed in hematopoietic stem cells and therefore responsible for reprogramming committed progenitors into leukemia stem cells (Krivtsov and Armstrong, 2007). The increased expression of TERRA in MLL-rearranged ALL is reminiscent of an increase in telomeric transcripts occurring as part of telomere reprogramming in induced pluripotent stem cells (Yehezkel et al., 2011). Upregulation of TERRA by MLL oncoproteins may be accountable for telomere reprogramming occurring within the transformation of committed progenitors towards lymphoblastic leukemia.

  1. Did they have any idea about TERRA modulation in ALL or AML patients?

We currently do not have data on TERRA modulation in MLL-rearranged or not leukemia patients. However, we are in the process of collaborating with colleagues from the Centro Ricerca M. Tettamanti – University of Milano Bicocca, Monza, Italy, directed by Dr. Andrea Biondi, with the intent to collect specimens from MLL-rearranged and non-MLL rearranged pediatric leukemia patients for the validation and extension of our findings on TERRA expression levels. This would be the objective of our next study.

  1. How could this information affect leukemia management? Could be hypothesized a target therapy counteracting the increased TERRA activity?

Further studies need to be achieved to confirm the sustained expression of TERRA in MLL-r ALL cell lines as part of a transcriptional signature that may help explain the genomic stability previously found in MLL-r pediatric ALL. For example, studies need to be undertaken to elucidate whether sustained levels of TERRA in MLL-r ALL lead to more rapid telomeric chromatin response to activation of DNA repair pathways, which may favor telomere integrity, and, ultimately, genomic stability. Confirmatory results in this direction on leukemic patient samples will sustain the hypothesis of targeting TERRA as a new therapeutic approach with the intent to undermine the potential genomic stability associated with its sustained expression in MLL-r ALL. As requested by the reviewer, a brief sentence has been added at the end of the Discussion to emphasize these potential therapeutic implications.

Sincerely,

Corrado Caslini

Reviewer 3 Report

It would be very interesting if the authors make the molecular evaluation even on samples of patients

Author Response

We thank the reviewer for the thoughtful comment and interest in the molecular evaluation of TERRA levels on patient samples. For clarification, as correctly suggested by another reviewer, we are now changing the title of this article to reflect that the data in this study were collected entirely on hematopoietic cell lines. The new title is:

“Telomere transcription in MLL-rearranged leukemia cell lines: increased levels of TERRA associate with lymphoid lineage and are independent of telomere length and ploidy”

We are currently in the process of collaborating with colleagues from the Centro Ricerca M. Tettamanti – University of Milano Bicocca, Monza, Italy, directed by Dr. Andrea Biondi, with the intent to collect specimens from MLL-rearranged and non-MLL rearranged pediatric leukemia patients for the validation and extension of our findings on TERRA expression levels. This would be the objective of our next study.

Sincerely,

Corrado Caslini